# The Potential of Satellite Imagery for Surveying Whales

**DOI:** 10.3390/s21030963

**Published:** 2021-02-01

**Authors:** Caroline Höschle, Hannah C. Cubaynes, Penny J. Clarke, Grant Humphries, Alex Borowicz

**Affiliations:** 1BioConsult SH GmbH & Co.KG, Schobüller Str. 36, 25813 Husum, Germany; 2British Antarctic Survey, High Cross, Madingley Road, Cambridge CB3 0ET, UK; penny@ukaht.org; 3Scott Polar Research Institute, Department of Geography, University of Cambridge, Cambridge CB2 1TN, UK; 4UK Antarctic Heritage Trust, High Cross, Madingley Road, Cambridge CB3 0ET, UK; clarke.pennyj@gmail.com; 5HiDef, Aerial Surveying Limited, 17 Silvermills Court, Edinburgh EH3 5DG, UK; Grant.Humphries@hidefsurveying.co.uk; 6Department of Ecology & Evolution, Stony Brook University, New York, NY 11794, USA; alexjborowicz@gmail.com

**Keywords:** very high-resolution (VHR) satellite imagery, remote sensing, great whale species

## Abstract

The emergence of very high-resolution (VHR) satellite imagery (less than 1 m spatial resolution) is creating new opportunities within the fields of ecology and conservation biology. The advancement of sub-meter resolution imagery has provided greater confidence in the detection and identification of features on the ground, broadening the realm of possible research questions. To date, VHR imagery studies have largely focused on terrestrial environments; however, there has been incremental progress in the last two decades for using this technology to detect cetaceans. With advances in computational power and sensor resolution, the feasibility of broad-scale VHR ocean surveys using VHR satellite imagery with automated detection and classification processes has increased. Initial attempts at automated surveys are showing promising results, but further development is necessary to ensure reliability. Here we discuss the future directions in which VHR satellite imagery might be used to address urgent questions in whale conservation. We highlight the current challenges to automated detection and to extending the use of this technology to all oceans and various whale species. To achieve basin-scale marine surveys, currently not feasible with any traditional surveying methods (including boat-based and aerial surveys), future research requires a collaborative effort between biology, computation science, and engineering to overcome the present challenges to this platform’s use.

## 1. Introduction

The study of wildlife using satellite imagery commenced with the Landsat-1 satellite (80 m spatial resolution), initially launched in 1972 to monitor land cover, and later extended for habitat surveys and to map the distribution range of some species [1,2,3]. Since then, there have been several technical developments, including improvements of optical sensors for an increased spatial resolution, with the development of very high-resolution (VHR) satellite. VHR satellites offer a sub-meter spatial resolution, allowing the detection of individual animals such as cattle [4], large African animals [5,6], polar bears (*Ursus maritimus*) [7], grey seals (*Halichoerus grypus*) [8], and albatrosses (Family: Diomedeidae) [9]. 

Monitoring whales using VHR satellite imagery was first attempted using the IKONOS-2 satellite in 2002; however, the 0.82 m spatial resolution provided by this satellite was not sufficient to make confident identifications [10]. A change in US legislation in August 2014 increased the maximum spatial resolution for commercial satellite operators to 0.25 m. Rather than coarse imagery with rudimentary outlines of large objects, the greater spatial resolution of 0.25 m imagery allows the capture of more detail where smaller features can be identified with greater confidence. A subsequent survey by Fretwell and colleagues [11] identified and counted southern right whales (*Eubalaena australis*) in an image captured by the WorldView-2 satellite (0.46 m spatial resolution). The available spatial resolution at the time, did not allow for the identification of whale-defining features—such as the fluke. The launch of the WorldView-3 satellite in 2016, offering the highest commercially available spatial resolution of 0.31 m, is capable of capturing whale-defining features [12], increasing the confidence in detections and making this satellite the most adapted for whale surveys. 

Traditional cetacean surveys are challenging and expensive, often requiring dedicated research vessels or aircraft and are thus limited in their temporal and spatial scope e.g., [13,14]. Much of the ocean’s surface has not been surveyed for whales [15,16], leaving large knowledge gaps [17,18] that hinder efforts to make sound conservation decisions that could more firmly protect threatened or endangered cetacean populations [19]. While cetacean surveys will be challenging with almost any survey platforms, VHR satellite imagery has the potential to fill knowledge gaps in whale research, allowing access to remote or rarely or infrequently monitored regions, as well as surveys of large spatial scope.

Building on recent work using VHR satellite imagery, and with the development of accessible and inexpensive artificial intelligence (AI) tools, there is a potential to (1) fill data gaps on the distribution, abundance, density, and population trends of whales; (2) supplement field-collected data by helping better plan aerial and boat-based surveys; and (3) allow for continued monitoring of whales when person-dependent surveys are restricted—as we have experienced during a pandemic—building a mechanism for important conservation research to continue globally. However, this method is in its infancy, with further pilot studies and experimentation needed. Here we aim to provide an overview of the development of these methods so far, in order to provide advice on future directions to follow and the main challenges that remain.

Because of the growing interest in using VHR satellite imagery as a new platform to study marine mammals, it is important to facilitate the discussion between stakeholders regarding the best approaches to data collection and analysis. Therefore, in December 2019, we led an international workshop entitled: “Marine mammal surveys from satellite imagery: Applications, automation, and challenges” during the World Marine Mammal Conference in Barcelona. A group of 33 participants from academia, consultancy, government, and industry came together to discuss the most recent advancements in satellite technologies and image recognition for marine surveys and the future of this field of study.

## 2. Considerations and Challenges Inherent to Satellite Images

VHR satellite imagery can either be accessed through satellite tasking (i.e., image capture is requested at a specific time and place), or archives of previously captured images (Figure 1). Archival image libraries are limited in areas of open ocean, as the majority of tasking occurs when and where there is demand and is largely concentrated in terrestrial areas, e.g., [20,21]. When requesting an image to be captured, satellite providers require a time window of several weeks, due to prioritisation of governmental and military acquisitions. This is mostly true for coastal areas. 

Currently all VHR satellites are commercially owned (e.g., Maxar Technologies, Airbus, and Earth-i) with the exception of one governmentally-owned satellite (Cartosat-3 for India). As a result, acquiring VHR satellite imagery comes at a cost. When considering the cost of an aerial or ship-based survey in a remote region, the cost of imagery may be competitive [22], but costs certainly can be a barrier to surveying large areas of several thousands of km^2^. Ultimately image costs limit the potential to develop the application of VHR satellite imagery to study whales. Funding is limited in the field of marine conservation, and the current costs for acquiring imagery restrict the potential for investment by NGOs and research institutes, and for inclusivity and the opportunity of developing nations to utilise this platform. 

When images are captured, environmental conditions can affect the quality of the image and the likelihood of detecting whales. Only cloudless images with few or no white caps will be useful for whale research, as clouds conceal the surface of the ocean, including whales, and strong winds create white caps, a confounding feature, that can create confusion and limit the confident detection of whales [12]. It may be possible to overcome the challenges of cloud presence through the use of radar sensors, capable of capturing weather independent imagery. However, the use of radar is dependent upon future technological advances in spatial resolution. At present the spatial resolution of 1 m for radar sensors (e.g., the TerraSAR-X) is too coarse for the detection of whales. Imagery with a low swell should also be preferred, as swell is known to influence the capacity to confidently detect whales [23]. It is worth considering that sea conditions are likewise a confounding factor in all visual cetacean surveys, including boat-based and aerial surveys. The WorldView-3 satellite, similar to other VHR satellites, captures both high spatial-resolution panchromatic images (i.e., 0.31 m greyscale band) and high spectral-resolution multispectral images, which provides a range of eight colour bands for analysis at 1.24 m resolution. Combining the two images during the image pre-processing stage, known as pansharpening (Figure 1), results in a very high-resolution coloured image (i.e., eight colour bands at 0.31 m spatial resolution) that tends to be preferred when detecting whales in satellite imagery. The coastal blue band, one of the eight bands available with Worldview-3, penetrates deeper into the water column, offering new perspectives of identifying features below the surface. Fretwell and colleagues [11] utilised the coastal blue band, identifying more whale-like features, interpreted as sub-surface, than were identified using other bands, such as the panchromatic, and proved to be the most successful band when automatically detecting whale-like features using thresholding techniques. While the coastal blue band can be used to detect whales below the surface, it will likely only allow the observation of whales near the surface, as all light will be absorbed by seawater at a certain depth [24]. Despite its potential, there will be whales beyond the detectable range of coastal blue. This may not be a concern for estimating whale abundance using satellite imagery as detecting whales below the surface may not be necessary for the future use of satellite imagery to estimate abundance. The reason being, correction factors accounting for whales below the surface, similar to those used in abundance estimates derived from aerial and boat surveys [25,26], could be adapted to satellites [27]. 

Researchers have explored both manual [10,11,12,27] and automated [28,29] methods of analysis (Figure 1). Manually counting whales in satellite images is currently the most accurate method, although the most time-consuming, as it can require approximately 3 h and 20 min to scan 100 km^2^ [12], and be erroneous due to observer bias [30]. Developing automated approaches using image-recognition techniques, driven by deep learning algorithms appears to be the way forward, particularly for surveying large marine regions. The revolution in the number and accessibility of satellites and the advances of computational power have led to increased use and recognition of deep-learning methods for whale monitoring.

Automated approaches are inherently more efficient than manual scanning, simply because humans are not required during the detection stages freeing up researchers to focus on other tasks. However, effective automation requires careful construction of a dataset of training images, from which the model can learn the characteristics of a whale in various ocean images. The training dataset must include a wide array of example images of different species of whales, in different environmental conditions, and displaying different postures/behaviors, as well as features that could be mistaken for a whale (Figure 1). Given the current limits on the available data of labelled satellite images of whales and similarly labelled images of confounding features within satellite imagery (e.g., rocks, boats, and white caps), required to create such a training dataset, aerial imagery can be used as a substitute for satellite imagery, to get the process underway. These aerial images should be down-sampled to the spatial resolution of VHR satellite imagery [28]. Alternatively, freely available photo archives of aerial and satellite imagery, can be accessed from sources such as Google Earth [29]. However, the satellite images from such sources are formatted in a way that reduces the spectral resolution (i.e., only three colour bands (red, green, and blue), compared to eight colour bands for the WorldView-3), and geospatial information (i.e., a jpeg format instead of a tiff). Further progress will be made by repetitively re-training the model, by incorporating new images as the algorithm detects them.

With automated approaches it is expected that some whales will likely be missed (false negatives) or that non-whale features will be counted as whales (false positives). Ideally with the appropriate training dataset, an automated method would yield no false negatives and only a few false positives. In this scenario, all positive detections can be manually checked and the process can improve over time by incorporating new observations into the existing training data (Figure 1). Another option is to quantify the false negatives and positives and account for them in biological analysis. For the time being, automated processes will likely necessitate a semi-automated approach, assisted by manual review. However, the overall technique has the potential to dramatically improve how quickly we can assess the distribution and population of some large cetaceans, as long as we can differentiate species, which is currently not feasible as VHR satellites has only been used in areas of known homogenous species presence. 

## 3. Future Directions

With the continued development of high-performance computing infrastructure and the planned launch of 10 new 0.30 m resolution satellites by Maxar Technologies (https://www.maxar.com/splash/it-takes-a-legion) and Airbus, by 2021 (https://www.airbus.com/space/earth-observation/portfolio.html), VHR satellite imagery is on its way to becoming a valuable tool to monitoring large whales, particularly in remote, understudied, and underfunded regions. Currently, we can reliably conduct presence/absence surveys through manual detection; however, continuing to develop automated detection systems will improve the efficiency. Throughout the development of the use of satellite imagery to monitor whales, we must ensure collaboration between various stakeholders (i.e., from industry, government, and academia), the creation of an open-source database of labelled whale images from satellites, and an appropriate analysis (coding) framework, as shown in Figure 1.This will provide consistent and powerful solutions for conservation. Future research should also focus on (1) building large and complex datasets to train automated systems; (2) pre-processing workflows to identify a suitable standard method; (3) species identification in order to conduct density, abundance, and population trend assessments; (4) testing the ability of models to detect whales across different sea conditions to allow transferability; and (5) understanding and accounting for factors limiting detectability such as weather conditions, including clouds and strong winds generating white caps [11,12]. 

Further experimentation and pilot studies will improve the reliability of this platform and help us understand how best to combine imagery methods with traditional field approaches to improve whale conservation. Future accessibility to VHR satellite imagery will dictate what can be achieved. To protect our world’s oceans, we need a global effort and we need to create opportunities for that to happen. Accessibility to VHR satellite imagery is critical to the development of expertise and to enhance conservation efforts.

## Figures and Tables

**Figure 1 sensors-21-00963-f001:**
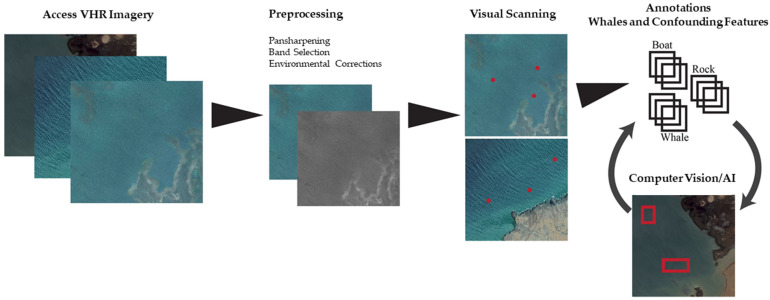
Processing steps to detect whales in very high-resolution (VHR) satellite imagery.

## Data Availability

Not applicable.

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
