# Peer review of "The Potential of Satellite Imagery for Surveying Whales"

_sensors, 2021, doi:10.3390/s21030963_

Round 1

Reviewer 1 Report

This is an interesting article and one that adds nicely to areas of ecology and conservation biology that requires further attention. It has been proved that very high-resolution (VHR) satellite imagery is helpful in the detection and identification of wildlife. Combined with efficient automated approaches, VHR satellite imagery may become a reliable and valuable platform to continuously monitor whales and other marine species. The work in the paper presents the development of relevant methods and discussed future research directions based on the main challenges inherent to satellite imagery. The specific comments are as follows:

There are few references cited in this paper, and part of citations are repeated. It is better that authors add more references to expand introduction and illustration of current challenges, which may provide more clear review of the field for readers in context of extensive literatures.

Authors provide some advises for future research, but they are like some preliminary ideas just in theory. What are the scientific innovations and comparative advantages of this paper compared to previous researches in relevant fields? Is there any limitation in this article?

On the basis of future directions that mentioned, how to build a feasible analysis framework (for surveying whales or other marine mammal) using VHR satellite imagery? Maybe this can be more specific.

Author Response

The authors would like to sincerely thank the reviewer for the careful review of the manuscript. The feedback and comments were greatly appreciated.

Each comment from the reviewer has been listed here in the same order of appearance as presented in the reviews. In the revised manuscript changes are also in blue.

  • Adding more reference to the introduction and challenge sections: we have added
    • Reference [6] of Duporge et al. 2020 on automatically detecting elephants from space L46
    • Reference [13, 14] Viquerat and Herr 2017 and Hamond et al. 2016 showing spatial and temporal limitations with traditional cetacean survey methods L61
    • Reference [17, 18] Redfern et al. 2017 and Mannocci et al. 2017 dealing with large knowledge gaps in data poor areas L62
    • Reference [19] Reynolds et al. 2009 on conservation in marine mammals in L64
    • Reference [21, 22] Brent et al 2020 and Xue et al 2017 on VHR use in terrestrial studies L89
    • Reference [25, 26] Buckland et al 2001 and Teilmann et al. 2013 on the use of correction factors L129
    • Reference [30] Bowler et al 2020 on the discrepancies between observers when counting wildlife from space L133.
  • What are the scientific innovations and comparative advantages of this paper; therefore, we have rewritten a few lines L41- 43 and added a few lines L146-152.
  • Adding a feasible analysis framework: we add a flowchart as Figure 1 and reference it at L87, L115, L131, L144, L159, L180.

Reviewer 2 Report

The manuscript aims to outline the perspective of extending research methods for whale populations using VHR satellite imagery.
Increasing the effectiveness and reducing the costs of surveying endangered species is an essential and topical subject. This Communication's strength is that the overview presented on undertaking surveys with VHR satellite imaging is comprehensive in organisational aspects. However, technically speaking, it appears the authors focus on a single aspect of detecting objects, namely whales. That is the interpretation of the animal's body parts appearing above the water. It is worth mentioning that with the use of suitable combinations of satellite bands, an opportunity might arise to identify the silhouette of a large object located shallowly under the water surface. For this purpose, near-blue radiation ranges which are used in coastal studies may be useful. It appears that including this aspect could assist in overcoming the uncertainties arising, due, for example, if the sea surface is slightly waved.
In future directions, the possibility of using registration techniques outside VHR sensors is also worth mentioning. In this respect, radar sensors can show potential, for example, Terrasar-X (Airbus), where the operator declares the spatial resolution of the SpotLight product up to 1m.

Specific Comment
Line 41: Here there is a misunderstanding that the Landsat satellite with its 80 m spatial resolution was launched for habitat surveys. To be precise, the aim was land monitoring and habitat surveys, but with regard to land cover. As the authors themselves point out in a later text, the animal population cannot be directly monitored with such a resolution.

Author Response

The authors would like to sincerely thank the reviewer for the careful review of the manuscript. The feedback and comments were greatly appreciated.

Each comment from the reviewer has been listed here in the same order of appearance as presented in the reviews. Our responses are shown in blue beneath each comment. In the revised manuscript changes are also in blue.

  • Mention of the use of the coastal blue band available with some VHR satellites was suggested; therefore we have added a few lines L117-129 discussing the potential use of the coastal blue band
  • Mention of the potential of radar sensors for the study of whales was suggested; therefore, we have added a few lines L105-108 discussing whether it could be applied for whales.
  • Line 41: Here there is a misunderstanding that the Landsat satellite with its 80 m spatial resolution was launched for habitat surveys. To be precise, the aim was land monitoring and habitat surveys, but with regard to land cover. As the authors themselves point out in a later text, the animal population cannot be directly monitored with such a resolution.

Line 41-42: We realise the ambiguity of this sentence and have rephrased it to convey our message more clearly, as we meant that Landsat-1 has been used for habitat surveys and to map distribution range, and not that Landsat-1 was built and launch for that purpose.

Reviewer 3 Report

In the abstract (line 20), I think you meant "< 1 m spatial resolution", instead of "> 1 m spatial resolution".

Author Response

The authors would like to sincerely thank the reviewer for the careful review of the manuscript. The feedback and comments were greatly appreciated.

Each comment from the reviewer has been listed here in the same order of appearance as presented in the reviews. Our responses are shown in blue beneath each comment. In the revised manuscript changes are also in blue.

  • Line 20: I think you meant "< 1 m spatial resolution", instead of "> 1 m spatial resolution".

Line 20-21: thank you both for spotting this error, we changed to “less than 1 m spatial resolution”.

Reviewer 4 Report

This is a brief paper about the opportunities of research whale populations by counting them in high resolution satellital images.

I just have one comment:

1. In the abstract:

(> 1 m spatial resolution)

(less than 1 m spatial resolution)

Author Response

The authors would like to sincerely thank the reviewer for the careful review of the manuscript. The feedback and comments were greatly appreciated.

Each comment from the reviewer has been listed here in the same order of appearance as presented in the reviews. Our responses are shown in blue beneath each comment. In the revised manuscript changes are also in blue.

  • Line 20: (> 1 m spatial resolution) (less than 1 m spatial resolution)

Line 20-21: thank you both for spotting this error, we changed to “less than 1 m spatial resolution”.